# A zero inflated log-normal model for inference of sparse microbial association networks

**Vincent Prost**[1,2]*, **Stéphane Gazut**[2], **Thomas Brüls**[1]*

**1** Génomique Métabolique, Genoscope, Institut François Jacob, CEA, CNRS, Univ Evry, Université Paris-Saclay, Evry, France, **2** Université Paris-Saclay, CEA, List, Palaiseau, France

* prost@genoscope.cns.fr (VP); bruls@genoscope.cns.fr (TB)

**Data Availability Statement:** All relevant data are within the manuscript and its Supporting information files.

**Funding:** V.P was supported by a Ph.D grant from CEA's High Commissioner office ("Thèse Phare").

## Abstract

The advent of high-throughput metagenomic sequencing has prompted the development of efficient taxonomic profiling methods allowing to measure the presence, abundance and phylogeny of organisms in a wide range of environmental samples. Multivariate sequence-derived abundance data further has the potential to enable inference of ecological associations between microbial populations, but several technical issues need to be accounted for, like the compositional nature of the data, its extreme sparsity and overdispersion, as well as the frequent need to operate in under-determined regimes.

The ecological network reconstruction problem is frequently cast into the paradigm of Gaussian Graphical Models (GGMs) for which efficient structure inference algorithms are available, like the graphical lasso and neighborhood selection. Unfortunately, GGMs or variants thereof can not properly account for the extremely sparse patterns occurring in real-world metagenomic taxonomic profiles. In particular, structural zeros (as opposed to sampling zeros) corresponding to true absences of biological signals fail to be properly handled by most statistical methods.

We present here a zero-inflated log-normal graphical model (available at https://github.com/vincentprost/Zi-LN) specifically aimed at handling such "biological" zeros, and demonstrate significant performance gains over state-of-the-art statistical methods for the inference of microbial association networks, with most notable gains obtained when analyzing taxonomic profiles displaying sparsity levels on par with real-world metagenomic datasets.

## Author summary

The importance of associations in the structuring and dynamics of community members is widely acknowledged, but we are currently unable to co-culture most of the micro-organims sampled from the environment. Computational methods to predict microbial associations can therefore be of practical interest, in particular given the large amounts of multivariate microbial abundance data generated by metagenomics. This data can in theory be leveraged to infer association networks, but with limited success so far, as several of its attributes lead to technical difficulties, including its extreme sparsity, compositionality and overdispersion among others. In particular, structural zeros (as opposed to sampling

The funders had no role in study design, data collection and analysis, decision to publish, or preparation of the manuscript

**Competing interests:** The authors have declared that no competing interests exist.

and technical zeros) corresponding to true absences of biological signals frequently fail to be properly handled, and such non-random absences can lead to high levels of false positives. Given their prevalence, zero values should be properly handled by the modeling process by accounting for the zero generating process in the first place. We describe here a truncated log-normal graphical model that specifically addresses zeros originating from biological absences, and discuss consistent methods for estimating sparse and high-dimensional association networks. We also show that this model generates sparse multivariate counts more close to those derived from real-world microbiomes.

This is a *PLOS Computational Biology* Methods paper.

# 1 Introduction

Metagenomics has increased our awareness that microbes most often live in communities structured by both environmental factors and ecological associations between community members. Understanding the structure and dynamics of microbial communities thus requires to be able to detect such associations, and is of both fundamental [1] and practical importance as it may ultimately enable the design and engineering of consortia of interacting organisms in order to fulfill various needs (e.g. improving the efficiency of wastewater treatment plants [2, 3] and designing more robust fecal transplants [4]).

Interactions within these microbial systems can have a positive, negative or null impact on the involved organisms, leading to a typology of pairwise interactions based on combinations of these win or loss outcomes, e.g. Lidicker [5] distinguishes the mutualism (+/+), competition (-/-), predation or parasitism ((+/-) or (-/+)), amenalism ((0/-) or (-/0)) and commensalism ((+/0) or (0/+)) interaction types. However, because of the current inability to co-culture most of the microbes sampled from the environment [6, 7], computational methods play a key role in predicting microbial associations. The advent of high-throughput sequencing and metagenomics resulted in the production of large amounts of multivariate microbial abundance data that could in theory be leveraged to reconstruct microbial association networks [8]. In practice however, only limited success has been met in terms of robust structure inference of real-world microbial association networks, and different methods frequently yield quite different results [9].

In principle, methods that consider the conditional dependency structure of microbial networks (like probabilistic graphical models) should perform better than methods using only univariate associations to predict pairwise associations, because the former has the power to resolve direct from indirect associations (e.g. associations between two species mediated through a third species). For example, ref [10] showed that "univariate networks" (i.e. networks based on pairwise associations predicted from univariate statistical associations) include high proportions of false-positive predictions when there is a substantial level of dependence between the samples, while the number of false positives was highly reduced when conditional networks were computed.

In the framework of Gaussian Graphical Models (GGMs), the structure of the inverse covariance matrix $\Sigma^{-1}$ (also known as the precision matrix) encodes an undirected graph whose edges represent conditional dependencies between variables [11]. Powerful graph inference algorithms for the latter include the graphical lasso (glasso) [12] and neighborhood selection [13]. However and importantly, the Gaussian assumption does not accomodate the excess of zeros typical of metagenomic datasets.

Indeed, a key difficulty for the statistical methods is the extreme sparsity and overdispersion of real-world sequence-based abundance data, including non random absences (structural zeros) that can lead to high levels of false positive results. Early population-level metagenomic studies highlighted extreme sparsity patterns in the human microbiome, with a large proportion of taxa being rare and absent in the majority of subjects, resulting in far more zero counts for each taxon than expected on the basis of Poisson, negative binomial, or Dirichlet-multinomial distributions [14].

Given their prevalence, zero values need to be properly handled by the modeling approaches, which requires accounting for the zero generating process in the first place. Zero values are often considered resulting from either the stochastic nature of sampling (sampling zeros), failed measurements arising from technical bias (technical zeros) or zero values resulting from a true absence of signal (structural or "biological" zeros) [15]. Previous work addressing technical zeros (absence of data) include [16], while [17] is an example of recent work dealing with sampling zeros.

Most of the existing work focuses on Gaussian graphical models, which also includes Poisson log-normal models where counts follow Poisson distributions with parameters sampled from a latent multivariate gaussian variable (network inference proceeds in the latent space) as illustrated in [17–20]. Methods based on log-normal models in the GGM framework include gCoda [21], Banocc [22], and Gaussian copulas with zero-inflated marginals [23, 24]. Outside the realm of GGMs, previous work on Poisson graphical models include [16] and [25]. However, there is currently no consistent multivariate Poisson distribution to model dependencies between count variables, as Poisson graphical models fail to have proper joint distribution [25, 26] or to have both marginal and conditional Poisson distributions [27].

While there is an abundant literature about models developed for the reconstruction of ecological networks from microbial occurrence or abundance data, these typically make strong assumptions about the zero generating process (e.g. that all zeros are explained by a common probability model) and don't investigate the consequences of deviating from these assumptions. Yet, the classical distinction between sampling zeros and structural zeros is not refined enough to describe the assumptions underlying commonly used models, like zero-inflated ones, which can lead to substantial biases under both simulated and real-world data settings [15].

We present here a truncated log-normal graphical model for inferring microbial association networks that accounts for both the compositionality of microbial abundance data and specifically addresses zeros originating from biological absence (structural zeros), and discuss consistent methods for estimating sparse and high-dimensional inverse covariance matrices.

Regarding the generative aspect of the model, which is important for producing realistic abundance data (e.g. for benchmarking purposes), an attractive approach (implemented in the popular Spiec-Easi toolkit [23]) relies on Gaussian copulas coupled with count distribution marginals in order to generate multivariate count distributions. Dependencies between variables can be simulated with a latent multivariate Gaussian distribution, which is then composed with count distributions, typically zero-inflated negative binomials. This protocol yields multivariate zero-inflated negative binomial distributions, but it does not produce compositional data, nor does it generate realistic count distributions when the proportion of zeros and the variances are large. We propose a zero-inflated log normal model for generating sparse multivariate counts, and demonstrate that it produces more realistic counts (Fig 1).

## 2 Motivation of the proposed model

The key motivation to use a multivariate zero-inflated model with a latent multivariate normal variable to account for the dependency structure of the model is partly inspired by the work of Lee et al., in the context of variable selection [14], with the following specific motivations:

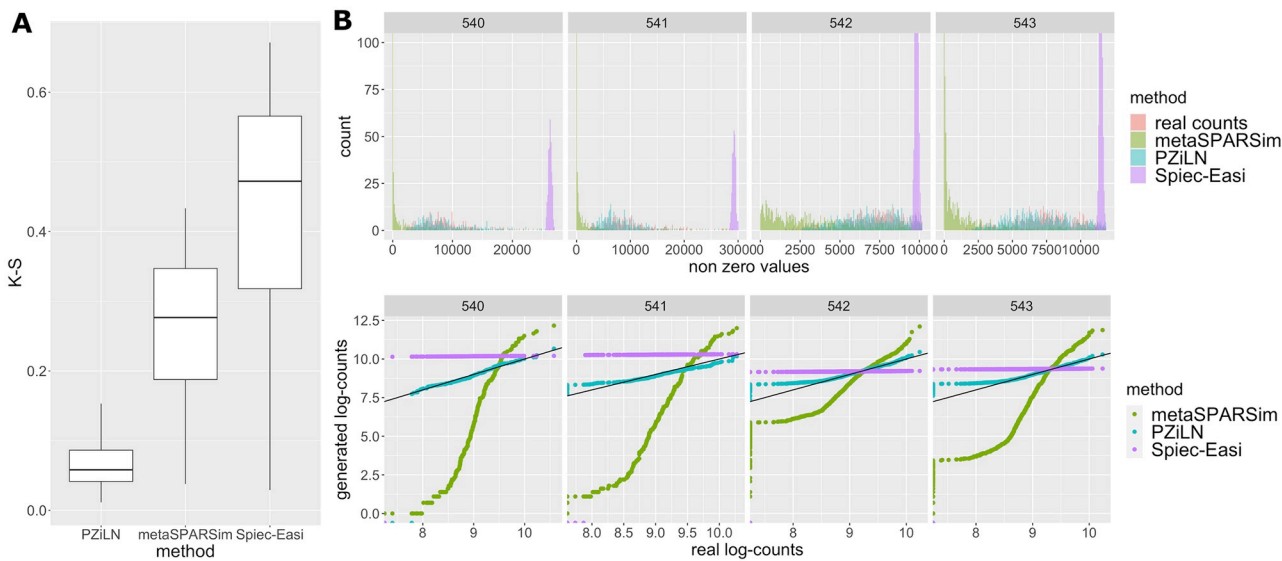

**Fig 1.** (A) Global feature by feature difference between the real (LifeLines-Deep cohort) and simulated counts measured by Kolmogorov-Smirnov statistics. (B) Top: histograms showing real and simulated count data. Real counts obtained from shotgun sequencing the microbiomes of the LifeLines-Deep cohort are shown in red; green, purple and blue represent synthetic counts generated respectively by metaSPARSim [28], spiec-easi [23] and our model (Poisson Zero Inflated Log Normal). The different panels show results for different randomly selected taxa. Down: quantile-quantile (QQ)-plots showing the logarithm of real counts vs the logarithm of synthetic counts.

- to have a better representation of real counts (e.g. improving over commonly used zero inflated negative binomial models) (Fig 1)

- to have a truly compositional data generator

- to take structural (biological) zeros into account at the network inference level

- to leverage the vast literature and experience available on log-normal models (e.g. [17, 18, 21], among others)

## 3 Materials and methods

We describe next the zero inflated log-normal model and a corresponding network inference method. Observed counts are noted $y_{ij}$ where $i = 1..n$ indexes the samples and $j = 1..p$ the variables (taxa). We denote the vector $y_i$ of size $p$ the observation of counts for sample $i$ and more generally in bold letters vectors of size $p$. The count vectors are gathered into a $n \times p$ matrix $Y$.

### 3.1 The model

Our model is derived from the multivariate Poisson log-normal model [29] to which a zero-inflated component is added. We consider a latent multivariate normal variable $z_i$ and a variable $a_i$ representing the real (unknown) abundances. We define the multivariate distribution of $a_i$ as following:

$$
\begin{aligned}
z_i \quad &\sim \mathcal{N}(\boldsymbol{\mu}, \Sigma) \\
a_{ij} \quad &= \mathbb{1}_{z_{ij} > \delta_j} e^{z_{ij}}
\end{aligned}
\tag{1}
$$

where $\mathcal{N}(\boldsymbol{\mu}, \Sigma)$ represents the multivariate normal distribution with mean $\boldsymbol{\mu}$ and covariance $\Sigma$ and $\mathbb{1}$ is the indicator function (with $\delta$ determining the zero probability).

In Gaussian Graphical Models, the structure of $\Sigma^{-1}$ encodes an undirected graph with edges representing conditional dependencies between variables [11]. In this model, zeros are interpreted as biological zeros and are related to a latent gaussian variable.

In metagenomics, the observed counts $y_i$ reflect the abundance proportions $\pi_i$ of microbes present in the sample: $\pi_{ij} = \frac{a_{ij}}{\sum_j a_{ij}}$. The number of reads observed for each sample is generally variable, and we will note it $N_i$. The counts can then be modeled with a Poisson or multinomial distribution whose parameter is proportional to $\pi_i$:

$$y_i \sim \mathcal{M}(N_i, \pi_i) \tag{2}$$

For example, [28] describes a method for simulating 16S rRNA gene count data based on gamma distributions, but it only moderately fits real data when sparsity and overdispersion are high, as shown in Fig 1. The same figure makes apparent that the zero-inflated negative binomial distribution is neither a good fit when parameters of the distribution are adjusted to real-world metagenomic data by using the data generation protocol of [23].

## 3.2 Network inference method

We next describe a network structure inference method suitable for metagenomic data consistent with the generative model presented above.

We will first present a centered log-ratio (clr)-like data transformation preserving scale invariance in the presence of zeros, and then propose an inference method for learning the sparse structure of $\Sigma^{-1}$ that is inspired by the work of [18].

**3.2.1 Data transformation.** Because of variations in sequencing depths across samples, the total number of reads in a sample only reflects the relative proportion of microbes it contains [30]. This compositional nature of sequence data prevents naive correlation-based analyses without appropriate correction [29], of which log-ratio based methods are the most commonly used. In the latter, the handling of zeros by adding a fixed pseudocount (i.e., irrespective of the sequencing depth of the various samples) has been shown to introduce important biases when the datasets include rare taxa and/or are inhomogeneous in coverage [10]. Downsampling or ignoring samples to level off sequence depth, or aggressive thresholding to remove rare taxa, can somewhat alleviate this bias, although this has been criticised as statistically unsound [31] and is performed at the cost of discarding large amounts of possibly biologically relevant data.

One way to address this difficulty, as originally shown by Aitchison [29] and implemented in popular toolkits (e.g. [23]), is to apply a centered-log-ratio (clr) transformation to the counts:

$$y_{ij}^{\mathrm{clr}} = \log(y_{ij}) - \frac{1}{p}\sum_{k=1}^{p}\log(y_{ik}) \tag{3}$$

An important property of this transformation is scale-invariance, which guarantees that two samples with similar read proportions will have similar transformed count data. Indeed, if counts represent absolute abundances with a scaling factor $a_i \approx \lambda y_i$, then we would have $a_i^{\mathrm{clr}} \approx y_i^{\mathrm{clr}}$.

As clr is not defined in zero, a typical workaround consists in adding a unit pseudocount to the original data in order to avoid numerical problems. However, doing so breaks the scale invariance property, as shown in S2 Text. We therefore propose a slightly different version of

clr, preserving scale invariance in the presence of zeros and defined as following:

$$\tilde{y}_{ij} = \begin{cases} \log(y_{ij}) - \frac{\sum_{k,\, y_{ik} \neq 0} \log(y_{ik})}{|k,\, y_{ik} \neq 0|} & \text{if } y_{ij} \neq 0 \\ 0 & \text{if } y_{ij} = 0 \end{cases} \tag{4}$$

where $|k, y_{ik} \neq 0|$ denotes the number of non zero entries of $\boldsymbol{y}_i$. A proof that this transformation preserves scale invariance in the presence of zeros can be found in S2 Text.

**3.2.2 Effect of clr transformation on the estimation of $\Sigma^{-1}$.** In "canonical" (non zero-inflated) log-normal models, when $\boldsymbol{a}_i \sim \mathcal{LN}(\boldsymbol{\mu}, \Sigma)$, we have $\boldsymbol{a}_i^{\text{clr}} \sim \mathcal{N}(F\boldsymbol{\mu}, F\Sigma F)$, where $F = I - \frac{1}{p}J$ and $J$ is a $p \times p$ matrix filled with ones. Even though there is no guarantee that $F\Sigma F$ is close to $\Sigma$ (in the sense of the Frobenius norm) when $p >> 0$, as shown in S1 Text, the approximation appears reasonably good in practice (see e.g. [23]).

In the zero inflated log-normal model, we will assume similarly that if $\boldsymbol{a}_i$ follows a distribution as defined in Eq 1, then $\tilde{\boldsymbol{a}}_i$ follows a zero inflated normal distribution defined as:

$$\begin{aligned} \boldsymbol{z}_i &\sim \mathcal{N}(\tilde{\boldsymbol{\mu}}, \tilde{\Sigma}) \\ \tilde{a}_{ij} &= \mathbb{1}_{z_{ij} > \tilde{\delta}_j} z_{ij} \end{aligned} \tag{5}$$

and assume that $\tilde{\Sigma}$ is a good approximation of $\Sigma$.

**3.2.3 Sparse network inference.** We will assume that $\tilde{\boldsymbol{y}}_i \approx \tilde{\boldsymbol{a}}_i$ follows the zero inflated normal distribution defined in Eq 5. For network structure inference, we follow a procedure similar in spirit to [18] and [24], with distinct data transformation steps to estimate the latent layer, whose key steps can be summarized as follows:

- obtain initial estimates of parameters $\hat{\mu}_j$, $\hat{\Sigma} = (\hat{\Sigma}_{11}, \hat{\Sigma}_{22}, \ldots, \hat{\Sigma}_{pp})$ and $\hat{\delta}_j$

- transform data to posterior mean of $Z|\tilde{Y}$ to obtain $\tilde{Z}$

- infer the structure of $\Sigma^{-1}$ with either the graphical lasso (glasso) [12] using the empirical correlation matrix of $\tilde{Z}$, or neighborhood search (Meinshausen and Bühlmann, MB) [13] on $\tilde{Z}$.

The glasso [12] solves a penalized likelihood maximization problem for the multivariate normal distribution, and Ambroise and Chiquet have shown (personal communication and S4 Text) that the MB algorithm solves a penalized pseudo-likelihood maximization problem. On the other hand, [18] showed that this last step is therefore the maximization step of an EM algorithm, the two first steps yielding the expectation step.

**3.2.4 Initial parameter estimates.** We describe here how to obtain initial estimates for the model parameters $\hat{\boldsymbol{\delta}}, \hat{\boldsymbol{\mu}}, \hat{\Sigma}$. As shown in [18], a diagonal estimation of $\Sigma$ can be made in order to reduce the computational burden of the method (note [32] provides an algorithm for maximizing the penalized likelihood without the diagonality assumption). Under the constraint that $\hat{\Sigma}$ is diagonal, all variables are independent and marginal parameters $\hat{\mu}_j$ and $\hat{\Sigma}_{jj}$ can be estimated separately for each variable. We have the following likelihood from the marginal distributions, which have both a continuous and a discrete part:

$$\mathcal{L}(\mu_j, \Sigma_{jj}, \delta_j | \tilde{\boldsymbol{y}}_i) = \prod_i \left( \mathbb{1}_{\tilde{y}_{ij}=0} \Phi_{\mu_j, \Sigma_{jj}}(\delta_j) + \mathbb{1}_{\tilde{y}_{ij} \neq 0} f_{\mu_j, \Sigma_{jj}}(\tilde{y}_{ij}) \right) \tag{6}$$

where $f_{\mu_j, \Sigma_{jj}}$ and $\Phi_{\mu_j, \Sigma_{jj}}$ are respectively the normal distribution and the cumulative normal distribution of mean $\mu_j$ and variance $\Sigma_{jj}$.

The solution for $\hat{\delta}_j$ is straightforward and independent of other parameters:

$$\hat{\delta}_j = \min_{i,\ y_{ij}\neq 0} \tilde{y}_{ij} \tag{7}$$

The other parameters can be obtained by maximizing the log-likelihood:

$$
\begin{aligned}
\hat{\mu}_j, \hat{\Sigma}_{jj} = \quad & \mathrm{argmax}_{\mu,\sigma^2} \sum_{i,\tilde{y}_{ij}=0} \log(\Phi_{\mu,\sigma^2}(\hat{\delta}_j)) \\
& + \sum_{i,\tilde{y}_{ij}\neq 0} \log(f_{\mu,\sigma^2}(\log(y_{ij})))
\end{aligned}
\tag{8}
$$

We will note $\hat{\Sigma} = \mathrm{diag}(\hat{\Sigma}_{11}, \hat{\Sigma}_{22}, \ldots, \hat{\Sigma}_{pp})$.

**3.2.5 Posterior mean transformation.** Following [18], we use the algorithm on the inferred latent Gaussian variable $\tilde{z}_i$. $\tilde{Z}$ is obtained by posterior mean transformation:

$$\tilde{Z} := \mathbb{E}_{\hat{\boldsymbol{\delta}},\hat{\boldsymbol{\mu}},\hat{\Sigma}}[Z|\tilde{Y}] \tag{9}$$

As $\hat{\Sigma}$ is diagonal, each value of $\tilde{Z}$ can be computed separately (see S3 Text),

$$\tilde{z}_{ij} = \mathbb{E}_{\hat{\delta}_j,\hat{\mu}_j,\hat{\Sigma}_{jj}}[z_{ij}|\tilde{y}_{ij}] \tag{10}$$

for which the computation is easy:

$$
\tilde{z}_{ij} =
\begin{cases}
\dfrac{\int_{-\infty}^{\hat{\delta}_j} y.f_{\hat{\mu}_j,\hat{\sigma}_j}(y)dy}{\Phi_{\hat{\mu}_j,\hat{\sigma}_j}(\hat{\delta}_j)} & \text{if } \tilde{y}_{ij} = 0 \\[2ex]
\tilde{y}_{ij} & \text{if } \tilde{y}_{ij} \neq 0
\end{cases}
\tag{11}
$$

Interestingly, we noticed that fixing a small inaccuracy in [18] during data transformation at the level of the diagonal terms of $\tilde{S}$ in Equation (8) of S3 Text did not increase the support of inferred networks. Our correction actually leads to increased variance of individual observations with less covariance information being shifted across variables, therefore possibly leading to poorer support for the network structure.

## 3.3 Synthetic datasets

In order to compare the performance of various methods without knowing the true underlying associations among taxa in real datasets, we designed synthetic datasets with a controlled ground truth.

For our simulations, we tested two generative models. The first is the model described in section 3.1 with Eqs (1) and (2) (ZiLN). The second is the model implemented in [23] using Gaussian copulas and zero-inflated negative binomial marginals (combined in the NorTA protocol). The covariance matrix $\Sigma$ associated to a given graph topology is generated using the Spiec-Easi framework [23], with condition number $\kappa = 100$ and the number of edges $e$ equal to the number of variables $p$ (the sparsity assumption of the association graph is translated into the scaling of the number of taxon-taxon associations with the number of taxa).

**3.3.1 Synthetic networks for Simulation 1.** In the first simulation, we used the ZiLN generative model in conjunction with different network topologies (band, Erdos-Renyi and scale-free), and varying sample numbers $n$ = 50, 100, 300, 1000, 2000 and taxa numbers $p$ = 100, 200, 500. The library sizes $N_i$ were drawn from negative binomial distributions

$N_i \sim \mathcal{NB}(\text{mean} = 1.5 \times 10^6, \text{size} = 5)$. The sparsity levels of taxa $s_j$ (i.e. the expected proportion of zeros) was drawn from a uniform distribution $s_j \sim \text{unif}(0, 0.9)$, bringing the total number of zeros to 45% of count values. The correspondence to the parameter $\boldsymbol{\delta}$ is given by: $\delta_j = \Phi_{\mu_j, \sigma_j^2}^{-1}(s_j)$, and the mean of the Gaussian distribution was also chosen randomly from a uniform distribution: $\mu_j \sim \text{unif}(0, 3)$.

**3.3.2 Synthetic networks for Simulation 2.** In the second simulation, the sparsity level of the variables is varied while $p$ and $n$ are fixed to $p = 300$ and $n = 100$ respectively. We used a deterministic formulation for $\boldsymbol{s}$: $s_j = \frac{j^t}{d}$ where $t$ is chosen so that the mean of $s_j$ is equal to the target global proportion of zeros which can take values from 0, 10, 50, 70 and 90%. The parameters $\boldsymbol{\mu}$ and $N_i$ are chosen as in Simulation 1.

For this second simulation, we also used the NorTA framework from [23] for comparative purposes, with a latent normal distribution $\boldsymbol{z}_i \sim \mathcal{N}(0, \Sigma)$ and count values generated by composition with marginal inverse cumulative distribution functions (cdf) $\boldsymbol{y}_i = (F_1^{-1}(\Phi_{0, \Sigma_{11}}(z_{i1})), F_2^{-1}(\Phi_{0, \Sigma_{22}}(z_{i2})),, F_p^{-1}(\Phi_{0, \Sigma_{pp}}(z_{ip})))$, where $F_j$ are cdf of zero-inflated negative binomial distributions. As parameters for those distributions, we took means as $\text{mean}_j = e^{\mu_j}$, a size $\nu$ fixed to $\nu = 10$, and a probability of zero equal to $s_j$.

# 4 Results

## 4.1 Accounting for sparsity and overdispersion

In order to assess the ability of our model to deal with the high levels of sparsity and overdispersion that are inherent to natural microbial abundance datasets, we compared the extent to which synthetic count data generated under our model and two other model-based approaches are able to reproduce abundances of taxonomic profiles generated from human gut microbiomes of the population-based LifeLines-Deep cohort [33]. This cohort includes 1,135 individuals (474 men and 661 women) from the general Dutch population, whose gut microbiomes were shotgun sequenced using the Illumina short-read technology, generating an average of 32 million reads per sample (EBI dataset EGAD00001001991). We phylogenetically profiled the sequences from the LifeLines-Deep cohort using the metapalette software [34], and several samples by taxa matrices corresponding to different taxonomic ranks were constructed from these results, e.g. the raw species-level matrix contains 3957 taxa (variables) and about 90% of zero entries.

The performance of our model was compared to two other methods generating synthetic count data by integrating information from experimental count distributions in their model-based generative process: spiec-easi [23] and metasparsim [28]. Spiec-easi implements a Normal to Anything (NorTA) approach [35] to generate correlated count data by coupling Gaussian copulas with count distribution marginals (typically zero-inflated negative binomials) fitted on real-world count data [23], while metasparsim is a tool designed to generate 16S rRNA gene counts based on a two-step gamma multivariate hypergeometric model [28].

Fig 1 provides different views of the performance of the three methods, with the top panel of Fig 1B showing count histograms for different randomly selected taxa, while the bottom panel presents the same results in the form of quantile-quantile plots (QQ-plots). Both representations make apparent that the zero inflated log-normal model provides a better fit to the very sparse and overdispersed data extracted from the real-world microbiomes. In order to provide a more global assessment of the differences between real and simulated counts, we measured the feature by feature differences between these distributions with the Kolmogorov-Smirnov statistic (i.e. the largest absolute difference between the observed cdf of the real and

simulated taxa abundances), thus providing a distance between them. Fig 1A shows that the proposed method achieves a smaller per-feature difference with the real distribution.

This observation also hints that state of the art methods (Spiec-Easi [23] and metaSPARSim [28]) used to simulate multivariate count data from rRNA amplicons fail to properly account for counts derived from shotgun metagenomic data, probably because of higher overdispersion in the latter.

## 4.2 Accuracy of microbial network inference

We compared our method against four state of the art methods for inferring microbial association networks. Magma [24] is a method for detecting microbial associations specifically aiming to deal with an excess of zero counts, while also taking compositionality and overdispersion into account; it is based on Gaussian copula models with marginals modeled with zero-inflated negative binomial generalized linear models, and the core algorithm for network structure inference is the graphical lasso, with its inference procedure relying on the estimation of the latent data by a specific medium imputation procedure [24]. Spiec-easi [23] is a popular and widely used toolkit for inferring microbial ecological networks; it is designed to deal with both compositionality and sparsity, and relies on algorithms for sparse neighborhood and inverse covariance selection (i.e. the graphical lasso) for network structure inference. SparCC is a relatively older but widely used method dealing with compositional data and measuring the linear relationship between log transformed abundances [36]. Flashweave [37] is a recently developed tool based on the HITON [38] constraint-based conditional independence solver and leverages the so-called local-to-global learning framework to infer directly associated neighborhoods of variables in large systems [37].

We performed a comparison of these three model-based methods in terms of their ability to accurately recover the structure of controlled networks of different dimensions, topologies and sampling regimes (described in the subsection 3.3.1 of the Material and Methods). The performance of the methods was quantified by the area under precision-recall curves (AUPR, which should be preferred over ROC-AUC on imbalanced datasets with a lot of true negatives like microbiome taxonomic data), and is displayed in Fig 2 for synthetic datasets encompassing different topologies (band, Erdos-Renyi, scale-free), dimensions ($p = 100, 300, 500$) and sample numbers ($n = 50, 100, 300, 1000, 2000$). We also initially measured these performances using AUROC (shown in S3 Fig), and assessed the precision of the methods on the 50 most confident edge predictions (shown in S2 Fig), both of which are largely consistent with the AUPR results shown in Fig 2.

In practice, this comparison process does not rely on the inference of a single graph but of a series of graphical models of varying sparsity (including the empty and complete network as extremes), which together constitute the solution path. Each of the decreasing values of the sparsity-controlling parameter in the solution path leads to a graph solution, including its set of edges. If we denote the latter by $E_i$ for graph solution $i$, we therefore have $E_1 \subset E_2 \subset E_i.. \subset E_n$, which provides a natural ordering of the edges in these sets, e.g. edges associated to elevated sparsity-controling parameters are more reliable than edges appearing ectopically under low sparsity settings. This provides a rationale for ordering the edges and computing AUPR values.

Besides the expected dependence of the results on network topology, dimension and sample number, this figure makes apparent significant performance gains of the proposed method, in particular for the recovery of networks endowed with scale-free topologies that were shown to be the most difficult to reconstruct accurately in previous studies [23]. It should be noted however that all methods show only limited accuracy for this network topology when operating in the under-determined regime, where the number of variables (taxa, p) exceeds the number of

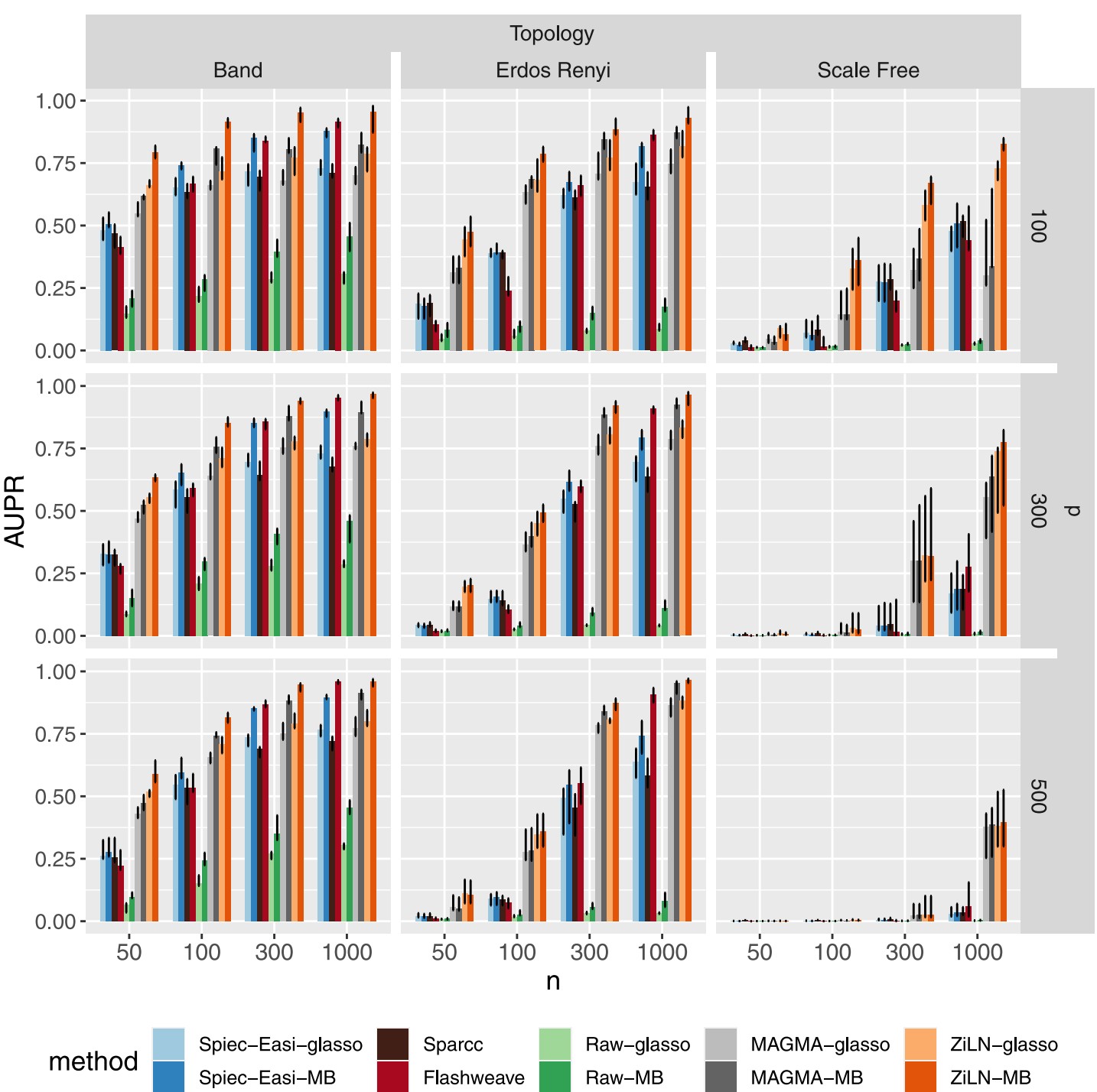

**Fig 2. Effect of the number of samples (*n*), number of taxa (*p*) and network topology on the performance of the various methods, measured with the area under precision-recall curve (AUPR), on synthetic datasets.** Bars represent the median over 10 runs, and error bars + 25% and −25% quantiles. Each method (Spiec-Easi (blue) [23], MAGMA (gray) [24] and ours (ZiLN, orange), as well as no transformation at all (green)), was tested with two structure inference algorithms (glasso and neighborhood selection). Two other orthogonal (i.e. based on distinct rationales) network inference methods, sparcc (dark-brown) [36] and Flashweave (dark-red) [37], were included to broaden the comparisons.

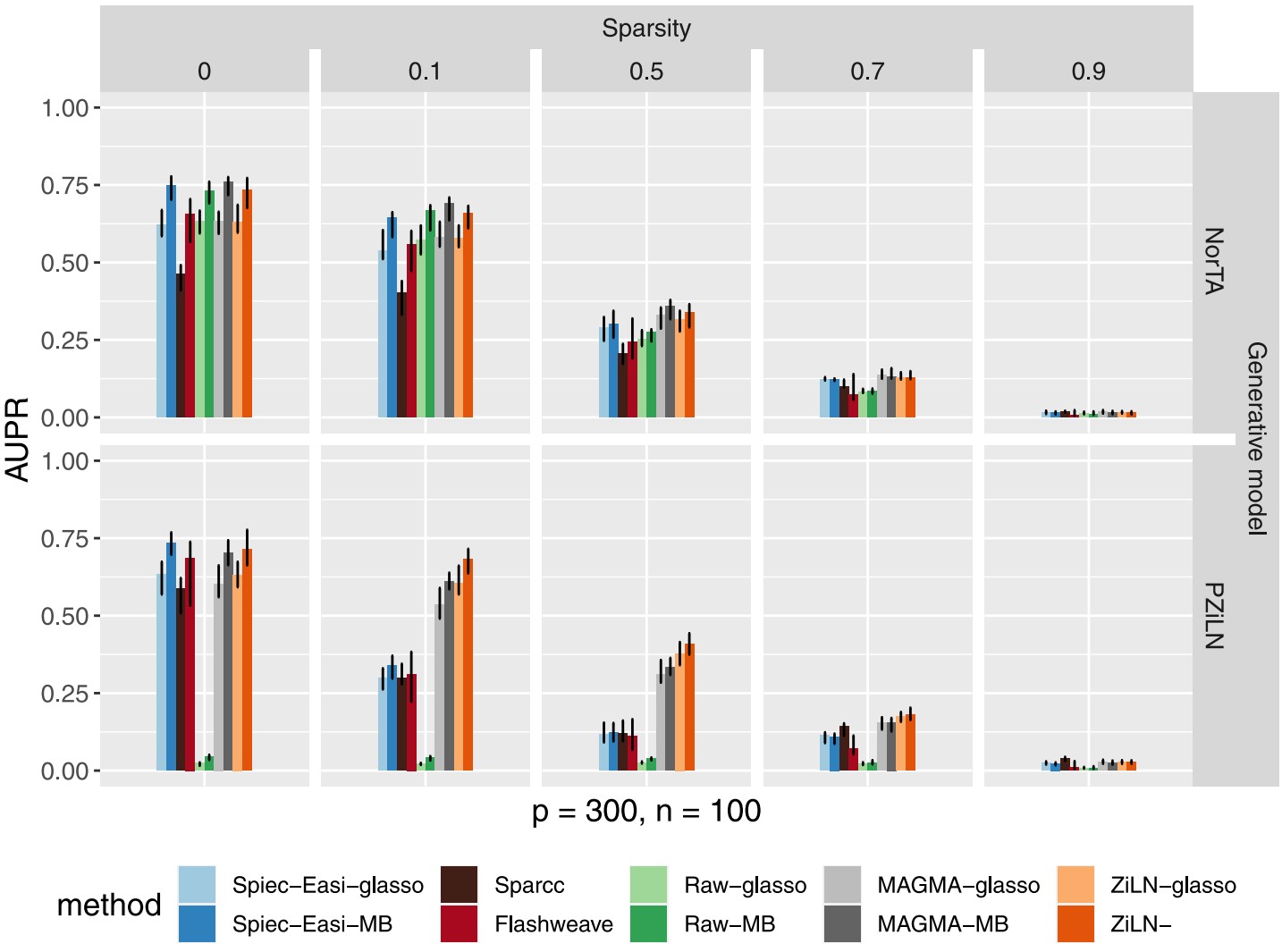

**Fig 3. Effect of the amount of zeros in the data (vertical panels) on method performance.** Top panel: network reconstruction accuracy for various methods using synthetic data generated with the NorTA protocol using zero inflated negative binomial marginals (see Methods). Bottom panel: network reconstruction accuracy using data generated under the zero inflated log-normal (ZiLN) model. Notations are as in Fig 2, and results are shown for the Erdos-Renyi topology.

samples (n). It should also be noted that the MB algorithm appears to perform better than the graphical lasso. A similar figure, presenting network reconstruction accuracy metrics for a distinct dataset generated using the NorTA protocol with zero inflated negative binomials, is shown in the Supplemental Information (see S1 Fig).

On the other hand, the experiments designed to probe the effects of increasing levels of sparsity (using the datasets described in the subsection 3.3.2 of the Material and Methods) make clear that all five methods are strongly affected by increasing proportions of zeros, as shown in Fig 3, with no method performing accurately under very high levels of sparsity (i.e., 90% of zeros).

## 4.3 Inference of real-world microbial association networks

We finally applied our method to infer microbial association networks from real-world taxonomic profiles generated from healthy gut microbiomes of the LifeLines-Deep [33] population cohort.

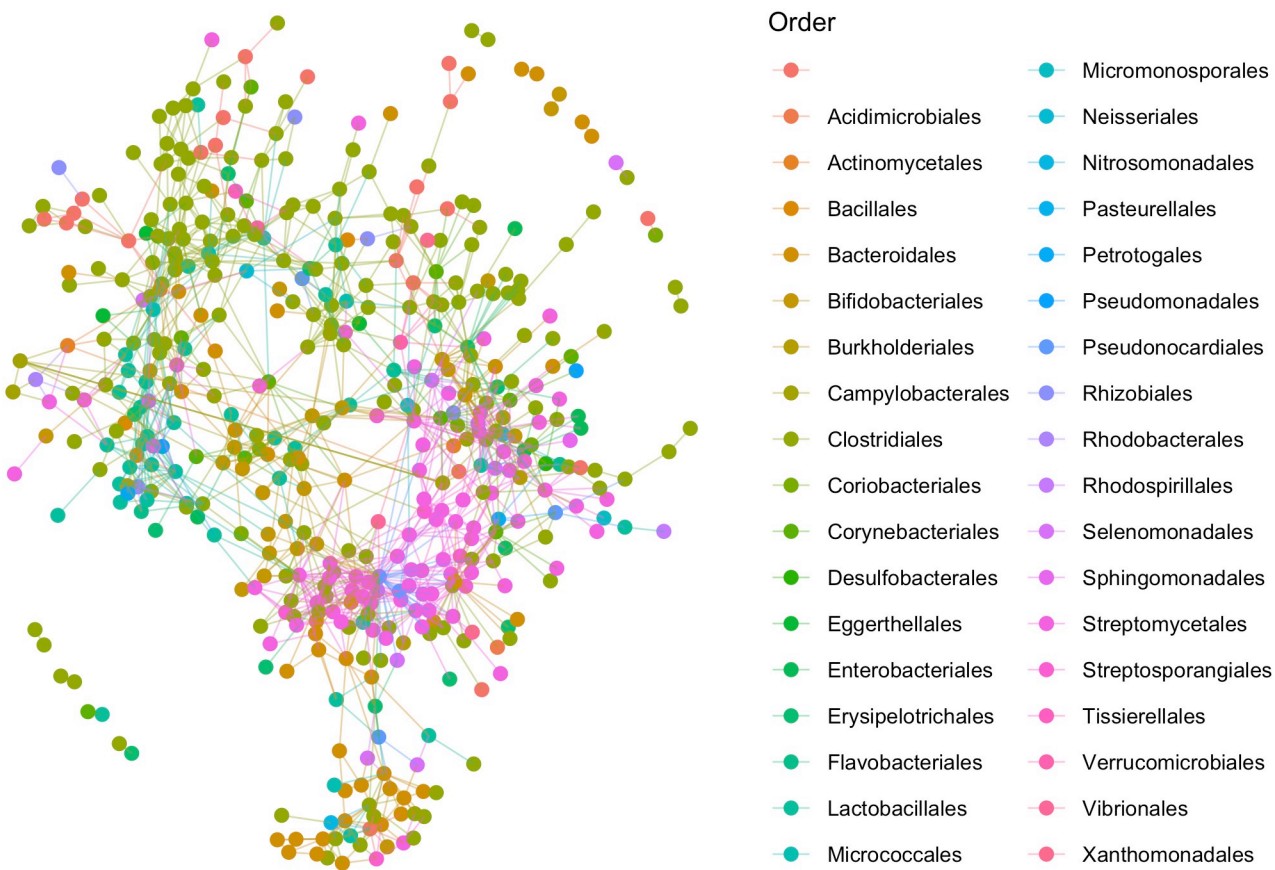

Order

| | |
|---|---|
| Acidimicrobiales | Micromonosporales |
| Actinomycetales | Neisseriales |
| Bacillales | Nitrosomonadales |
| Bacteroidales | Pasteurellales |
| Bifidobacteriales | Petrotogales |
| Burkholderiales | Pseudomonadales |
| Campylobacterales | Pseudonocardiales |
| Clostridiales | Rhizobiales |
| Coriobacteriales | Rhodobacterales |
| Corynebacteriales | Rhodospirillales |
| Desulfobacterales | Selenomonadales |
| Eggerthellales | Sphingomonadales |
| Enterobacteriales | Streptomycetales |
| Erysipelotrichales | Streptosporangiales |
| Flavobacteriales | Tissierellales |
| Lactobacillales | Verrucomicrobiales |
| Micrococcales | Vibrionales |
| | Xanthomonadales |

**Fig 4. Species-level association network inferred from the LifeLines-Deep microbiomes.** Nodes represent species and are colored according to their taxonomic order.

Even though confirmed ecological associations within natural microbial communities are only scarcely known, several global network properties have been reproducibly documented, including the salient observation of preferential associations among phylogenetically close organisms [39, 40]. This property, known as assortativity, is manifest at the level of highly connected components in our analysis, and is displayed for the species taxonomic level in Fig 4.

To provide a more quantitative support for the observed assortativity phenomenon, we computed assortativity coefficients (homophyly of the graph) for the different methods at various phylogenetic levels using the igraph R package [41] (shown in S1 Table).

In order to assess the extent to which the taxa association predictions are shared among the five methods, we measured the amount of common edges predicted by the different methods. In order to get comparable edge sets predicted by the different methods, we decided to pick the 1200 top edges for each method (e.g. this entails using a sparsity-controlling parameter for glasso yielding 1200 detected edges). Fig 5 illustrates the overlap between the different edge sets, and makes apparent the substantial and still problematic differences that exist among the various predictions. On one hand, this is partly expected and consistent with previous observations [9]. On the other hand, the figure shows that our method has the highest ratio (239/100) of edges predicted unanimously among all methods versus predictions unique to the individual methods.

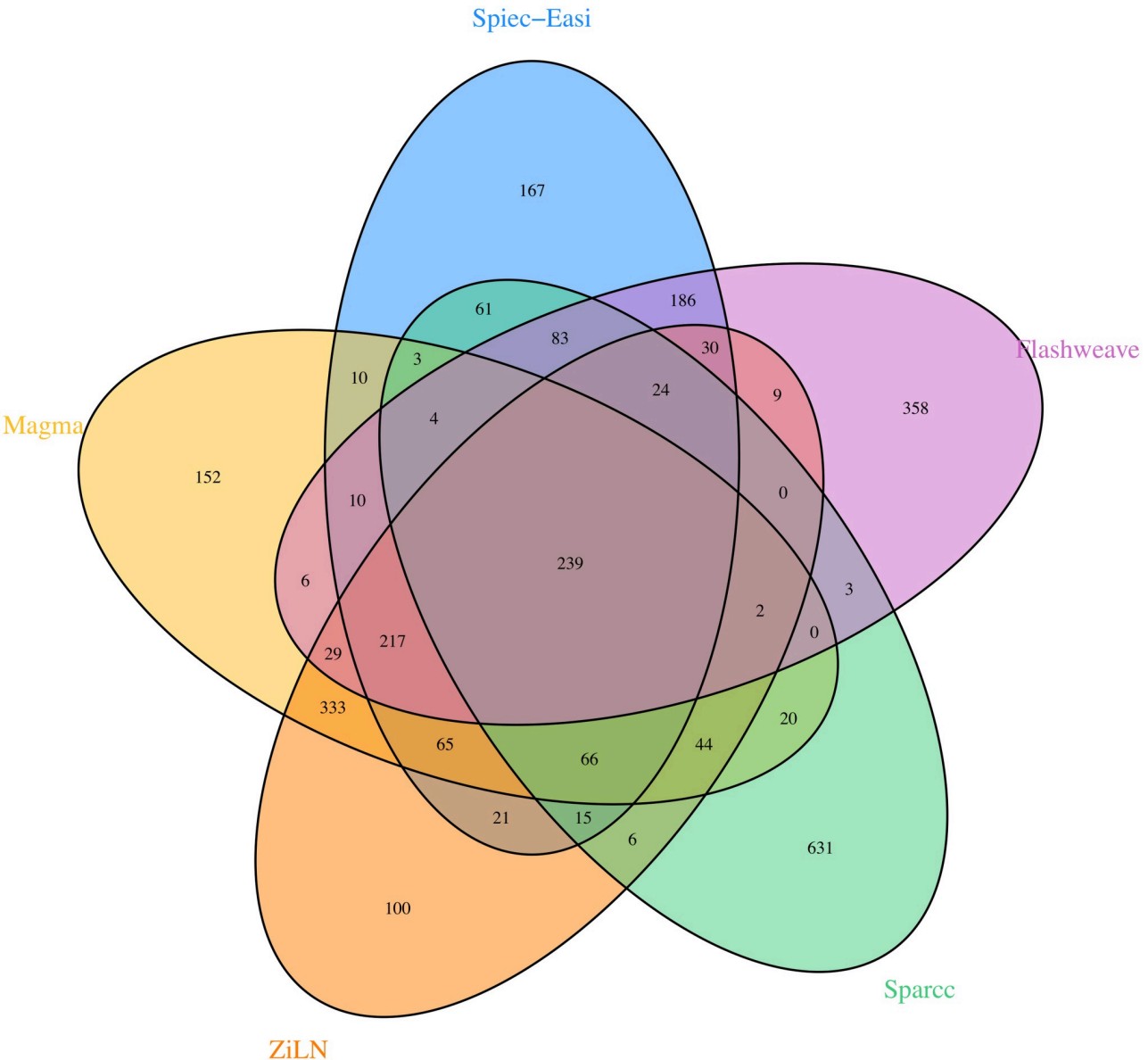

**Fig 5. Venn diagrams displaying the overlap between the species associations sets predicted by the different methods on the human gut taxonomic profiles of the LifeLines-Deep cohort [33].** To simplify the plot, a threshold was applied here to remove taxa absent in more than 20% of the samples, leading to a total of 565 species. It can be seen that our method has the highest ratio (239/100) of edges predicted by all methods versus idiosyncratic ones, but that overall substantial differences remain among the predictions of the various methods.

## 5 Discussion

Accurate inference of microbial association networks is necessary in order to gain a deeper understanding of the functioning of microbial consortia and to take advantage of the increasingly large body of data that is generated worldwide through environmental genomics initiatives. The formalism of Gaussian graphical models naturally lends itself to the problem of network reconstruction, but despite attractive mathematical attributes and the availability of powerful inference algorithms, application of this framework to the identification of robust microbial associations has only been met with moderate success. It appears unlikely though

that the problem lies in the GGM formalism itself or in inherently flawed inference algorithms; instead, it is most likely rooted in difficulties to properly account for the idiosyncrasies of biological data. Beyond compositionality and overdispersion, high levels of sparsity are common attributes of real-world biological datasets, which are challenging to account for in a statistically sound manner. In particular, an important distinction between sampling zeros and structural zeros is often neglected, and nearly all of the actual analyses of multidimensional metagenomic count data perform some form of aggressive thresholding to discard rare but possibly biologically relevant data.

We presented here a simple statistical model specifically aimed at accounting for large amounts of structural (biological) zeros in the network inference process, and provide evidence of its usefulness by showing performance gains with respect to several state of the art methods for microbial network inference. Unoptimized code as well as scripts to replicate the figures can be accessed at https://github.com/vincentprost/Zi-LN.

## Supporting information

**S1 Text. Problem with compositionality under the gaussian assumption.** Discussion of the issue with compositionality under the log-normal model in a high dimensional setting. (PDF)

**S2 Text. Problem with the clr transformation when there is an excess of zeros.** Discussion of the effect of the clr transformation in the presence of an excess of biological zeros. (PDF)

**S3 Text. One step EM.** Description of the one step EM procedure. (PDF)

**S4 Text. Neighborhood selection as a penalized maximization problem.** Formalization of neighborhood selection as a penalized maximization problem, justifying its inclusion in the one step EM procedure. (PDF)

**S1 Table. Assortativity coefficients.** Assortativity coefficients ($p < 10^{-4}$) of graphs for the different methods at various phylogenetic levels. (PDF)

**S1 Fig. Effect of the number of samples ($n$), number of taxa ($p$) and network topology on the performance of the various methods, measured with the area under precision-recall curve (AUPR), on synthetic datasets generated with the NorTA approach using zero inflated negative binomial marginals (see Methods).** Bars represent the median over 10 runs, and error bars +25% and −25% quantiles. Each method (Spiec-Easi (blue) [23], MAGMA (gray) [24] and ours (ZiLN, orange), as well as no transformation at all (green)), was tested with two structure inference algorithms (glasso and neighborhood selection). SparCC (dark-brown) [36] and Flashweave (dark-red) [37] are two unrelated inference methods based on a distinct (orthogonal) rationale, and were included for broadening the comparisons. (PDF)

**S2 Fig. Precision of the different methods on the top 50 edges only (could be compared to Fig 2, see main text).** (PDF)

**S3 Fig. This figure is analogous to Fig 2 in the main text, but with AUROC computed instead of AUPR.**
(PDF)

## Acknowledgments

We would like to thank Christophe Ambroise and Julien Chiquet for helpful discussions and advices, and Sawsan Kanj for generating the taxonomic profiles of the LifeLines-Deep cohort.

## Author Contributions

**Conceptualization:** Vincent Prost, Thomas Brüls.

**Data curation:** Vincent Prost, Thomas Brüls.

**Formal analysis:** Vincent Prost, Thomas Brüls.

**Funding acquisition:** Stéphane Gazut, Thomas Brüls.

**Investigation:** Vincent Prost, Stéphane Gazut, Thomas Brüls.

**Methodology:** Vincent Prost, Thomas Brüls.

**Project administration:** Stéphane Gazut, Thomas Brüls.

**Resources:** Vincent Prost, Thomas Brüls.

**Software:** Vincent Prost.

**Supervision:** Stéphane Gazut, Thomas Brüls.

**Validation:** Vincent Prost, Thomas Brüls.

**Visualization:** Vincent Prost.

**Writing – original draft:** Vincent Prost, Thomas Brüls.

**Writing – review & editing:** Vincent Prost, Thomas Brüls.

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
