## [Decision Letter · Decision Letter 0]

22 Mar 2021

Dear Dr. Prost,

Thank you very much for submitting your manuscript "A zero inflated log-normal model for inference of sparse microbial association networks" for consideration at PLOS Computational Biology.

As with all papers reviewed by the journal, your manuscript was reviewed by members of the editorial board and by several independent reviewers. In light of the reviews (below this email), we would like to invite the resubmission of a significantly-revised version that takes into account the reviewers' comments.

We apologise for the delay in review as one of the reviewers did not provide a review despite repeated reminders.

We cannot make any decision about publication until we have seen the revised manuscript and your response to the reviewers' comments. Your revised manuscript is also likely to be sent to reviewers for further evaluation.

Sincerely,

Niranjan Nagarajan

Associate Editor

PLOS Computational Biology

Stefano Allesina

Deputy Editor

PLOS Computational Biology

**Comments to the Authors:**

Reviewer #1: In this paper titled “A zero inflated log-normal model for inference of sparse microbial association networks”, Prost et al introduces a new algorithm for inferring association from microbiome data, taking into account of the data sparseness using a modified version of centered-log-ratio transformation. While this algorithm would likely contribute to the association inference toolbox, I do have some concerns on the methods and results.

Major concerns:

1. The authors tended to use “association” and “ecological interaction” interchangeably. As it is noted in recent works (Carr et al, 2019; Hirano & Takemoto, 2019), associations between microbial abundances do not necessarily mean interactions.

2. Introduction Line 76: How to tell a zero is a structural zero or a sampling zero? In practice, sampling zeros are essentially treated the same as structural zeros by this algorithm? Then how would the sequencing depth affect the performance of the algorithm?

3. Figure 1: as this figure only shows some examples, the authors might consider calculate a distance between simulated and true distributions, and show a comparison of the distances across all taxa.

4. In synthetic dataset 1 and 2, what was the number of edges in each simulation? Will this (i.e. the network density) affect the performance of the inference?

5. While AUPR is good for measuring the performance in some perspectives, it would be great if the authors could consider other metrics. For instance, the precision and recall (and F1 maybe) of the top 50 predictions is probably more important for practical usage.

6. Fig 4: It would be helpful to run some network clustering algorithm or community detection algorithm to demonstrate that phylogenetically related species were more often in the same cluster/community than random.

Minor concerns:

1. What is the \\delta_j in equation 1?

2. Introduction Line 9: I guess the authors mean “mutualism” (positive-positive interaction) instead of “commensalism” (neutral-positive interaction). Why are these four types of interactions “most important”?

3. Section 3.2.1: The first paragraph seems largely repetitive with the text in the introduction

4. Methods: How was the EM algorithm stopped?

Reviewer #2: This paper presents a truncated log-normal model to evaluate networks for microbiome data that accounts for compositionality and the zero-inflated counts. This model can be used to make inferences about microbial interaction networks and to simulate microbiome data. The authors compare simulated data results and application to a motivating dataset across different approaches. The paper is very well written and the model description is clear. I offer a few suggestions or points that might be helpful to clarify below.

1) How are the covariances between taxa affects by the 0 counts? Would rare taxa tend to have higher covariances simply because they tend to be absent from the majority of samples rather than due to an actual interaction? In other words, for two rare taxa that do not co-occur in any samples, is their covariance greater than 0?

2) The authors present the clr transformation to account for the compositionality of the abundance data, there are other transformations (ilr) that are equally common. I was wondering if the authors could comment on whether their approach can similarly be applied to the other transformations or if some of the attributes are specific to using the clr?

3) I don’t quite understand how the real counts in Fig 1 are aligned with the simulated counts, more specifically how are the real counts from taxa 540 matched up with the simulated counts from each approach. Did the authors use some information about the taxa from the real data to simulate counts for the same taxa? Or is there actually no pairing of information within each panel and the figure is meant to represent a range of distributions?

4) In Section 3.3.1, it wasn’t clear to me why the size for the negative binomial random number generator equaled 5. Same question for the parameters used to select the mean of the Gaussian distribution. Were these based on estimates from the real dataset?

5) In the results, the authors refer to methods being compared using the reference numbers (line 265), it would be helpful to also include the name of the method as it appears in the referenced figure so the reader doesn’t have to refer to earlier text.

6) I don’t have as much experience with the network approaches as other parts of the model, so my apologies if this is a well-known concept in the network literature. I don’t understand what information is being used to estimate the AUPR, the authors mention it is a measure of accuracy of network structure. Does it focus more on recovering the edges assuming all the nodes are the same or is that true only in this context since the nodes don’t change. A little bit more explanation or a reference would be helpful.

7) It would also be helpful to have some additional information for the real-world example related to the sparsity of the counts, a description of the average number of sequences per sample and the total number of taxa. This would help the reader put this data in context with the simulated data presented earlier in the paper.

8) Do the authors have an idea of how the results presented in Figure 5 might change using different values to define edges? It wasn’t clear how the presence of an edge was defined and whether this was consistently used for all approaches.

**Have all data underlying the figures and results presented in the manuscript been provided?**

Reviewer #1: Yes

Reviewer #2: Yes

PLOS authors have the option to publish the peer review history of their article (what does this mean?). If published, this will include your full peer review and any attached files.

Reviewer #1: No

Reviewer #2: No
---

## [Decision Letter · Decision Letter 1]

17 May 2021

Dear Dr. Prost,

We are pleased to inform you that your manuscript 'A zero inflated log-normal model for inference of sparse microbial association networks' has been provisionally accepted for publication in PLOS Computational Biology.

Best regards,

Niranjan Nagarajan

Associate Editor

PLOS Computational Biology

Stefano Allesina

Deputy Editor

PLOS Computational Biology

Reviewer's Responses to Questions

**Comments to the Authors:**

Reviewer #1: The authors have addressed all my concerns.

Reviewer #2: The authors have adequately addressed my previous comments. I appreciate the detailed responses, after reflecting more on my first comment regarding the covariance of two rare taxa, I believe that the authors modified approach to calculating the CLR completely alleviates my concern.

The authors also note from their simulations that data with 90% zeros were not described well by any method, this might be included as a consideration for the section related to the motivating example since that dataset also contained 90% zeros.

Neither of these comments are meant to suggest changes to the current version of the paper.

**Have the authors made all data and (if applicable) computational code underlying the findings in their manuscript fully available?**

Reviewer #1: Yes

Reviewer #2: Yes

PLOS authors have the option to publish the peer review history of their article (what does this mean?). If published, this will include your full peer review and any attached files.

Reviewer #1: No

Reviewer #2: No

---

## [Editor Report · Acceptance letter]

14 Jun 2021

PCOMPBIOL-D-20-02287R1 

A zero inflated log-normal model for inference of sparse microbial association networks

Dear Dr Prost,

I am pleased to inform you that your manuscript has been formally accepted for publication in PLOS Computational Biology. Your manuscript is now with our production department and you will be notified of the publication date in due course.

With kind regards,

Katalin Szabo
